# Is Probing All You Need?
# Indicator Tasks as an Alternative to Probing Embedding Spaces

**Tal Levy**
Bar-Ilan University
talevy17@gmail.com

**Omer Goldman**
Bar-Ilan University
omer.goldman@gmail.com

**Reut Tsarfaty**
Bar-Ilan University
reut.tsarfaty@biu.ac.il

## Abstract

The ability to identify and control different kinds of linguistic information encoded in vector representations of words has many use cases, especially for explainability and bias removal. This is usually done via a set of simple classification tasks, termed *probes*, to evaluate the information encoded in the embedding space. However, the involvement of a trainable classifier leads to entanglement between the probe's results and the classifier's nature. As a result, contemporary works on probing include tasks that do not involve training of auxiliary models. In this work we introduce the term *indicator tasks* for non-trainable tasks which are used to query embedding spaces for the existence of certain properties, and claim that this kind of tasks may point to a direction opposite to probes, and that this contradiction complicates the decision on whether a property exists in an embedding space. We demonstrate our claims with two test cases, one dealing with gender debiasing and another with the erasure of morphological information from embedding spaces. We show that the application of a suitable indicator provides a more accurate picture of the information captured and removed compared to probes. We thus conclude that indicator tasks should be implemented and taken into consideration when eliciting information from embedded representations.

## 1 Introduction

Pre-trained vector representations of words have introduced a stark improvement in the performances of any NLP task they have been applied to, and proved to be effective in classification and prediction of real world tasks (Peters et al., 2018; Devlin et al., 2018; Liu et al., 2019b; Zeng et al., 2021). This has led researchers to grow interest in analyzing the encoded linguistic features in the representations. To this end, probing tasks, initially proposed by Köhn (2016) and Adi et al. (2016), are frequently used.

Probing tasks (henceforth *probes*) are designed to evaluate the information existing in representations by training a simple classification model (Conneau et al., 2018; Niven and Kao, 2019). The existence of a probing model that can classify the queried linguistic property is assumed to imply that the respective linguistic information exists in the representations (Alain and Bengio, 2017; Belinkov and Glass, 2019).

However, probes have several, frequently contemplated, drawbacks. Generally, probes are meant to 'un-black-box' word representations, that is, to act as lenses into what information exists in the representations. Yet the actual relation between the probe's results and the representations thereof remains unclear. This has led some researchers to claim that the probing classifiers should be kept simple (Conneau et al., 2018), others are in favor of complex ones (Pimentel et al., 2020a), and a few advocate for dispensing with probing classifiers altogether (Immer et al., 2022).

A solution frequent in the literature (Wu et al., 2020; Zhou and Srikumar, 2022; Ravfogel et al., 2022, inter alia) is to accompany the results of probes that involve training an auxiliary classification model, with results over substantially different kind of tasks that are designed to query linguistic information directly from the representations, and do not require training another model. In this work we propose to collectively refer to such tasks by the name *indicator tasks*, as they provide direct indication of the stored information. We claim that indicator tasks are generally superior to trainable probing models, and provide more faithful information regarding the complex structure of the embedding spaces.

We substantiate our claim with two test cases that involves concept erasure: gender debiasing (Bolukbasi et al., 2016) and morphological property removal. We show that in both cases the evaluation of erasures with indicators provides a fuller

picture of the information erased and retained. Our two test cases are dissimilar in many ways. Gender debaising involves a single, binary, unwanted, social bias that models learn implicitly from the distribution of words in the corpus. Morphological features, on the other hand, are a series of multi-valued grammatical properties, expressed explicitly in the text, and are not undesired per se, yet their erasure may benefit processing tasks in morphologically rich languages.

We apply two different concept erasure methods to them, respectively: RLACE (Ravfogel et al., 2022) for gender debiasing, and INLP (Ravfogel et al., 2020) expanded to multiple features for morphological features removal. We evaluate the success of these methods with standard linear probes, as well as with suitable indicator tasks: Word Embedding Association Test (WEAT, Caliskan et al., 2017) and KNN-bias correlation (Gonen and Goldberg, 2019) for gender debiasing, and for morphological information removal we present a novel Double Edged Outlier Detection (DEOD) that indicates on both morphological and lexical semantic information. All representations are obtained from contextualized BERT models (Devlin et al., 2018).

The results in both test cases, despite their dissimilarities, point to the same direction. They show that the conclusions pointed to by the indicators are contradictory to those obtained from the probes. When removing gender bias from the representations we found probes to report complete erasure of the bias while our indicators still manage to pick it up from the vectors. In the second test case, we compared two possible extensions of INLP and found that probes point to one extension possibility as superior, while the indicator points to the other, not only by indicating the removal of the morphological properties, but also by examining the overlapping dimension of semantics.

We therefore conclude that when eliciting linguistic information from embedded spaces, results from indicator tasks should be presented alongside standard probes, and results from probes should not be overstated.

## 2 Background: The Probing Conundrum

The question of evaluating linguistic properties in dense word representations has been a long standing one, entering alongside the introduction of static word embeddings (Mikolov et al., 2013). Nowadays, probing tasks are generally accepted as the primary method for querying linguistic information in dense vectors. A *probe* is a classifier designed to elicit a property of interest directly from the representation (Adi et al., 2016; Alain and Bengio, 2017). Their simplicity and speed are appealing as opposed to the alternative of comparing performance over downstream tasks. However, researchers are yet to reach a consensus regarding several problems with probes (Immer et al., 2022).

Many issues with probes are derived from their usage of linear or neural models. Most prominently are the two interleaved problems of classifier and task selection. It is well received that a probing *classifier* should be simple (Liu et al., 2019a), since as Hewitt and Liang (2019) stressed, with enough training data, a sufficiently expressive probe could solve any task regardless of the representations. On the other hand, Pimentel et al. (2020a) have shown that simple tasks are not informative enough when applied to contextual representations. Thus a probing *task* should not be simplistic. However, researchers found that complex tasks often require a complex *classifier* (Belinkov and Glass, 2019), which are not recommended, as mentioned above.

We are thus led to a three-way trade off between the complexity of the task, the complexity of the classifier, and the efficacy of the probe. Given that simple models do not fare well on complex tasks, we cannot prefer a simple probing model and a complex task simultaneously. However, opting for either a more complex model or a simpler task would limit the reliability and quality of information given by the probe on the existence of a property in the embedded space.

Previous works suggested various solutions to mend these problems. Hewitt and Liang (2019) suggested to examine probes on control tasks, in which labels are distributed randomly, in addition to testing the probes on the desired task. Probes are then considered successful only if they perform well on the target task while failing on the control task. Alternatively, Elazar et al. (2021) suggested amnesic probing, in which a probe is considered successful only if it is possible to erase the information from the representations and make the same probe fail. The problem with these methods is that they take a process with complex problems and only complicate the process further, making it even harder to determine what in the results of a given probe is ascribable to representations, to the probe, or to the suggested new mechanism.

Another line of works suggested amending the output of the probing process itself by adding description length (Voita and Titov, 2020), mutual information (Pimentel et al., 2020b) or Pareto hypervolume (Pimentel et al., 2020a) to provide an indication on the *ease of extraction* of properties rather than on their existence. These methods, while useful, embrace the aforementioned trade off and relinquish the attempt of solving it. We, on the other hand, advocate for a completely different approach that bypasses the entire debate.

In general, the probing conundrum seems to stem from the presence of a train set which makes it unclear whether the probe employs artifacts in the training data to solve the task or evaluates the encoded information itself. Therefore an alternative that does not require a training set and a trainable model is warranted.

## 3 Introducing: Indicators

As a result of these various issues with probing, previous works had suggested to elicit linguistic properties from embedding spaces without the involvement of a trainable auxiliary model. These studies mostly relied on the similarity between vectors, either between the representations themselves (Wu et al., 2020) or between attention weights (Htut et al., 2019). In addition, clustering was also taken as a possible task derived from vectors alone (Zhou and Srikumar, 2021). Here we propose to refer to the different kinds of such tasks with the umbrella term *indicator tasks*, as they indicate the presence or absence of a property in the representation without any intervention or entanglement of the task itself with the vectors it is supposed to study.

By virtue of not involving a trainable classifier, indicators cancel out many of the arguments against probes. The tasks can be complex but still fast to devise and cheap to implement and run, without a risk of over-fitting the training data — as there is none. Most importantly, there is no other model that interjects, ascribing the performance directly to the embedded representations, where the task is essentially performed in a zero-shot setting. All it takes is to design or select the indicator that suits your needs. In a sense, indicators can be perceived as a resurgence of intrinsic evaluation methods that were popular with static representations. These methods were typically formulated as word similarity tasks, either explicitly (Hill et al., 2015, inter alia) or implicitly (Levy and Goldberg, 2014).

Thus, in order to formulate the indicators in our case studies we draw inspiration from such tasks.

The advantages of indicators have led to their use in various works, usually to complement the picture painted by standard probes. For example, Zhou and Srikumar (2022) used "DirectProbe" (Zhou and Srikumar, 2021) as an indicator in addition to a 2-layered non-linear classifier to reinforce and explain their conclusion that fine-tuning BERT does not always improve performance. Similarly, Merchant et al. (2020) used an MLP-based probe in addition to an indicator named "structural probe" (Hewitt and Manning, 2019) to show that after fine-tuning, word representations keep the linguistic structure they learn during pre-training. They completed the picture by showing with the indicator task of representational similarity analysis (Kriegeskorte et al., 2008) that fine-tuning affects the top layers more than the bottom layers. Lastly, the work presenting the concept erasure method RLACE Ravfogel et al. (2022), used in our first test case below, demonstrated its effectiveness by utilizing both probes and indicators.

Here we demonstrate that despite having a shared goal, i.e., querying information from dense representations, probes and indicators may result in contradicting conclusions. The implication of such disharmonious results is that if we want a fuller picture of the information existing in the representations, probing results must be accompanied with those obtained by indicators.

As will be demonstrated in the test cases below, the specific modus operandi of trainable probes makes them less likely to detect signals that exist in the embedding space but deviate from the pattern they were made to spot. We will demonstrate that by pitting linear probes, that were selected due to their simplicity in accordance with the issues laid out in the previous section, against simple similarly-based indicators. We will show that this difference is significant enough to make indicator tasks detect gender bias and morphological information that are undetectable by a probe.

## 4 Concept Erasure

Concept erasure methods are aimed at removing information from word representations. Formally, their objective is, given a set of word representations $X = \{x_1, ... x_n \in \mathbb{R}^d\}$ and a corresponding property distributed over words $P = \{p_1, ... p_n\}$, to learn a function $r(\cdot)$ such that

$r(X) = \{r(x_1), ...r(x_n)\}$ are not predicative of $P$, but preserve as much information as possible from $X$ (Ravfogel et al., 2022).

However, there is no consensus on how to evaluate the success of the erasure procedure. The evaluation methods presented in previous works usually rely specifically on the nature of the erasure method and the nature of the target property. In addition, timely field tendencies or "trends" are also influential, with earlier works devising intrinsic evaluation methods (Bolukbasi et al., 2016; Zhao et al., 2018; Gonen and Goldberg, 2019) and more recent works utilizing probes (Ravfogel et al., 2020, 2022).

Our case studies, presented in detail in the following sections, systematically compare both evaluation methodologies, with an added effort of adjusting the former to *contextualized* embeddings. Each of these test cases involves a different erasure method, a different property to be erased, and, to evaluate the success of the erasure method, different indicator(s) in addition to probes. Common to both cases is the contradicting conclusions drawn from evaluating the erasure method with probes compared to its evaluation with indicators.

## 5 Case Study 1: Gender Debiasing

Of all unwanted social biases in word representations (e.g, Nadeem et al., 2021), gender bias is the most researched both as an end goal in and of itself, and as a test case for information removal. We adopt gender debiasing as a simple test case of removing a binary unwanted property from word representations, and we examine the efficacy of a recent state-of-the-art concept erasure method on contextualized embeddings.

### 5.1 RLACE

Relaxed Linear Adversarial Concept Erasure (RLACE, Ravfogel et al., 2022) is a post hoc adversarial method for removing a single binary property from an embedding space. Unlike other adversarial methods that incorporate changes to the model during training, this method operates on the embedded vectors after the training is complete. It aims to identify a linear subspace from the representation and remove it, by solving a linear minimax game and projecting the vectors using the resulted matrix.

**Examples for male biased words**
successful, general, earned, president, politician, engineer, captain, fatal
**Examples for female biased words**
aesthetic, looks, singer, lipstick, pink, smiled, skirts, pretty

Figure 1: Examples from the 100 most biased words in the unperturbed embedding space induced by BERT.

### 5.2 Experiments

We assess the success of RLACE in removing gender bias from representations obtained from BERT (Devlin et al., 2018).[1] We use a standard probe to classify words as masculine or feminine, and compare it to indicators from Gonen and Goldberg (2019) that are used to identify bias, and that were originally executed over non-contextualized embeddings.

**Data** To create the contextualized vocabulary we used GUM (Zeldes, 2017), an English universal dependencies dataset (Nivre et al., 2020). Note that every word has several representations, each for every context. As is customary in gender bias works, we excluded from the vocabulary words that are explicitly gendered, such as *man, queen* and the like, in order to examine only *unjustified* gender bias.[2]

**Tagging biased words** In order to apply the indicator tasks, presented below, we first need to tag a set of words that are gender biased in the unperturbed space. Following Bolukbasi et al. (2016), we tag words as gender biased if their vectors are similar to those of explicitly gendered words. In practice, this is done by projecting the vectors of all non-explicitly-gendered words onto a *"gender direction"*, calculated by a 1D-PCA on a sub space spanned by the difference vectors of several gendered word pairs, e.g., $\vec{she} - \vec{he}$, $\vec{woman} - \vec{man}$, etc. The words corresponding to the 10,000 vectors with the highest positive projection are tagged *feminine* while those with the highest negative projection are tagged as *masculine*. Figure 1 exemplifies some of the words that appear to be with the strongest bias.

---

[1]We used HuggingFace implemented bert-base-uncased (Wolf et al., 2020).

[2]In addition, we excluded sentences in which at least one word is explicitly gendered, since, in the contextualized embedding case, other words in a sentence could legitimately inherit gender-related properties from these gendered neighboring words.

The list of biased words is fixed throughout the experiments. That is to say, both probes and indicators were tasked with discerning between biased and unbiased words that were tagged only based on the original BERT representations.

**Probes** We train and test a linear classifier (Perceptron; Pedregosa et al., 2011) to predict *masculine* and *feminine* of the words in the list.

**Indicators** Our indicators for this tasks are the ones used by Gonen and Goldberg (2019), extended here to a contextualized setting.

One indicator is the Word Embedding Association Test (*WEAT*), introduced by Caliskan et al. (2017). It fixes a set of words related to stereotypically gendered occupations and traits, and tests whether a group of words suspected as biased has a tendency for uneven association with one of the fixed word sets. We experimented with the sets related to career (stereotypically male) and family (female).[3] The metric reported is the standard *WEAT's d*, see Caliskan et al. (2017) for details.

Another indicator, devised by Gonen and Goldberg (2019), is meant to detect *implicit* bias, i.e., the situation in which words are not more similar to explicitly gendered words, but they may be similar among themselves. For example, an erasure method may move the word "nurse" further from "woman", but its neighbors may still include high abundance of stereotypically gendered words like "receptionist". Therefore, the *KNN-bias correlation* indicator checks the Pearson correlation between the percentage of biased words in the 100 nearest neighbors of a target word and the bias in the target word itself, i.e., its projection onto the gender direction.

## 5.3 Results

Table 1 includes the results for both probes and indicators before and after the application of RLACE. It shows that the probe's classification accuracy drops from an almost perfect accuracy before the erasure to to 48.7% after it, faring a bit worse than a random guess. On the other hand, the indicators' drop in performance after the RLACE application is far less impressive, with minor drops of 0.04 in WEAT's $d$ and 0.037 drop in the KNN-bias' Pearson correlation. In absolute terms, the indicators

| Method | Original | RLACE |
|---|---|---|
| Probe (Accuracy) | 99.95 | **48.7** |
| WEAT ($d$) | **1.17** | **1.13** |
| KNN-Bias (Pearson) | **0.638** | **0.601** |

Table 1: Measuring gender bias in BERT's embedding space before and after RLACE projection. Both the probes and the indicators should not detect bias after the application of RLACE, so lower is better. While probes point to no bias at all, both indicators point only to a small drop.

results still detect a bias, as 0.601 is by no means a negligible correlation and a $d$ score that is above 0.8 is considered high (Caliskan et al., 2017). Both results are significant with $p$-values of 0.

When judging the success of RLACE, it is clear that the probe and the indicators lead to markedly different conclusions. While the linear probe indicates the successful removal of the bias, it is only able to tell whether there is a linear separation between the potentially biased words. The indicators, on the other hand, are able to detect biases that are expressed by means that are not necessarily linear.

## 5.4 Discussion

All in all, this test case demonstrates that indicators were able to expose information that linear probes could not. This should not surprise us as the probing model is quite simple with the ability to detect only a very specific kind of separation.

Most probably RLACE was able to erase linear gender bias since it was designed primarily for linear concept erasure. However, as mentioned in Section 2, using a more complex probe that could detect non-linear separation would have come at a price of opacity. The indicators, on the other hand, were able to surface biases that were expressed in non-linear terms while still being simple and transparent due to the lack an auxiliary model.

In the next section we present a more complicated test case that involves the erasure of multiple properties, and therefore includes multiple probes whose results are less easily interpretable.

## 6 Case Study 2: Morphological Properties

In our second test case, we aim for representations that are ignorant of morphological features, such as tense, person etc. Using such representations can make the treatment of morphologically rich languages with many inflections per lemma simplified to a great extent, with models for downstream tasks not needing to worry about unseen inflections

---

[3]Caliskan et al. (2017) have a couple of more fixed sets: art related words for female bias and mathematics and science for male. However, there was no bias detected with respect to these sets even before debiasing.

(Gong et al., 2018; Czarnowska et al., 2019). As success criteria for removing morphological knowledge from word representations, we utilize probes and a novel indicator.

To this end, we extend an existing concept erasure method, Iterative Nullspace Projection (INLP, Ravfogel et al., 2020), to accommodate the erasure of multiple properties. We propose two possible, and novel, extensions and test these alternatives for their efficacy using standard probes, and using a novel indicator task that we design, which indicates on the presence of morphological and lexical semantic information simultaneously.

## 6.1 Iterative Nullspace Projection

INLP is a linear concept erasure method, with the objective that no linear classifier could do better than guess. The method constructs a subspace ignorant of a property $P$ (e.g tense) by iteratively training linear classifiers predicting $p \in P$ (e.g past) and projecting the representations unto their nullspace. The process sets to eliminate the entries used to predict the features by various classifiers. For more details see (Ravfogel et al., 2020). While INLP is designed for a single property, here we would like to use it for a complete feature bundle.

**Multi-Class Setting** INLP is intended to remove a single property $P$ which can correspond to any morphological property we choose. However, we would like to erase as many properties as possible. We therefore extend INLP by iterating over a set of morphological properties by applying INLP for each property in a procedure we term *Iterative Iterative Nullspace Projection* or *I²NLP*.

Although intuitive, extending INLP in this manner could damage the embeddings beyond usability, as each removed property reduces the dimensionality rank of the vectors. Moreover, repeated multiplications could be numerically unstable. Therefore we explore two possible variants of I²NLP.

In the first variant, *Regressive*, the representations are projected after each property removal, before moving on to remove the next property. In the *Non-Regressive* variant, the representations are only projected after a projection matrix is calculated for all properties, at the end of the procedure.

**Choosing a Variant** At first glance, the difference between the variants may seem small, but manipulating the input representations after each iteration, as done in the regressive variant, could affect the removal of the next properties significantly.

The representations could encode properties using overlapping entries, which upon removal in prior iterations, cannot be relied on. This fact also leads this variant to be order dependant — it is unclear how would the removal of property $P_1$ affect the removal of property $P_2$ and vise-versa. The non-regressive variant, on the other hand, is indifferent to order, but multiplying the projection matrices one after another could be numerically unstable.

In order to decide between the variants we test both of them using both a probe and an indicator task we designed, presented in the next section.

## 6.2 The Morpho-Semantic Indicator

Consider the quadruple (walked, hiked, strolling, bumped). Semantically, the words (walked, hiked, strolling) are related and (bumped) is the *semantic outlier*. Morphologically, the words (walked, hiked, bumped) are related as they share the -ed inflectional form, and (strolling) gerund is the *morphological outlier*. We follow this general scheme of quadruples with both a morphological and a semantic outliers to create a novel indicator task: *double edged outlier detection* (DEOD).

Outlier detection is an intrinsic method proposed by Camacho-Collados and Navigli (2016) with the objective of identifying a single outlier in a set of words. In our case the set contains two outliers, one semantic and the other morphological. By predicting a single outlier from each set, models take part in a zero sum game, and we get to evaluate which of the two dimensions of meaning is prioritized in the representations.

In the case of morphological erasure, this indicator task is also used to verify that the representations keep as much of the lexical semantic information as possible. It is important to note that attempting a similar approach with probes, i.e., eliciting information of the interplay between two partially overlapping dimensions of word-level information, is tough to envision.

**Task Definition** Given a quadruple $Q = \{w_1, w_2, w_3, w_4\}$, the task is to identify (one of) the outliers in a procedure based on cosine similarity. Our working hypothesis is that embedding spaces are capable to encode both lexical semantics and morphology to some extent and the interplay between them is affecting the decisions taken by the model when picking an outlier.

We use the two metrics termed *hard* and *soft* outlier detection, proposed by Camacho-Collados and

Navigli (2016). Hard outlier detection is measured with accuracy. It is the percentage of quadruples in which the semantic (morphological) outlier was chosen. Soft outlier detection, measured with Outlier Position Percentage (OPP), is based on the position of the true outlier in the ranked quadruple and not only reflecting whether it was ranked first. See Camacho-Collados and Navigli (2016) for further details about these metrics.

**Data Generation**  In order to generate data for the DEOD task we used UniMorph (Batsuren et al., 2022) to determine what words are of the same morphological category, and FastText embeddings (Bojanowski et al., 2017), to assess semantic similarity between words.[4]

Specifically, from UniMorph we sampled a word with a certain morphological characterization $T$, e.g., *walked* which is characterized as a past tense form. Then, using the word embeddings, we add the 2 most similar words with the same morphological characterization, for example *hiked* and *strolled*, and one dissimilar word as the semantic outlier, like *bumped*. The morphological outlier is then created by inflecting one of the semantically similar words to another morphological category, for example we could replace *strolled* with *strolling*. The result is the quadruple (walked, hiked, strolling, bumped). This procedure is repeated until enough quadruples are sampled. Since this method is language agnostic, it can be applied for different languages.

All in all, we created 2,000 quadruples for each language. Since both UniMorph and FastText cover many languages, the data generation method is easily applicable to many languages.

### 6.3 Experiments

**The Languages**  We experimented with three languages: English, Spanish and Hebrew, these languages are typologically different both in the number of morphological features they express and in the means they employ to express them; with one isolating, one suffixing and one templatic language.

**Probes**  In order to ascribe morphological information to words in their context we used the morphological layer of Universal Dependencies (UD; Nivre et al., 2020) as our data set.[5] This data was used both for determining what morphological properties are to be erased with $I^2$NLP and for training the probes over the final projected representations. The calculation of projection matrices as part of $I^2$NLP is done only using the train and dev sets, and the reported scores of the final probes are calculated over a test set of sentences that were not seen during training.

The linear probing model for each morphological category is a multi-class version of an SGD linear classifier (Pedregosa et al., 2011).

**Language Models**  For the contextualized representations we used the following models:

- Hebrew: AlephBERTGimmel (Guetta et al., 2022)
- English: bert-base-uncased (HuggingFace implementation; Wolf et al., 2020)
- Spanish: Beto (Cañete et al., 2020)

Since the indicator requires words to be represented out of context, a single representation per word was obtained by averaging the contextualized representations of all its occurrences in the UD sentences.

**Metrics**  For evaluating the $I^2$NLP variants using probes we report the classifier F1 score averaged over all morphological properties. We expect classifiers to perform poorly on representations that were obtained by a good $I^2$NLP variant. The evaluation of the $I^2$NLP variants using the DEOD indicator task is done using the task's own metrics: accuracy for hard outlier detection and OPP for soft outlier detection. We expect a successful $I^2$NLP to prevent the detection of the morphological outlier. On the other hand, we expect higher performance in detecting the semantic outlier.

**Baseline and Skyline**  For both the probe and the indicator we provide as a baseline the respective metric assessed over the original representation, without application of any $I^2$NLP variant.

As a skyline we evaluate the respective metric as it would be in an ideal situation, where all morphological information is completely erased from the representations. For the probe we provide the averaged F1 expected from a classifier that randomly guesses the morphological label. For the indicator we replaced all word vectors with those of the respective lemmas to ensure erasure of all morphological information.

---

[4]Static representations are used instead of contextualized ones since we need for the DEOD task a representations of complete words out of context.

[5]In cases where UD breaks white-spaced words to multiple units we merged the morphological tags of all subwords.

| Lang | Before | Reg | Non-Reg | Skyline |
|------|--------|-----|---------|---------|
| Heb | 94.52 | **61.89** | 79.62 | 40.28 |
| Spa | 92.99 | **50.32** | 76.88 | 33.89 |
| Eng | 92.80 | **46.39** | 64.24 | 37.5 |

Table 2: Before and after projection in average F1 scores on classifying Tense, Gender, Number, Person, Case and VerbForm. The better I²NLP variant prevents classification so lower is better. The regressive variant appears as the better method.

| Lang | Morphological | | | |
|------|--------|-----|---------|---------|
| | Before | Reg | Non-Reg | Skyline |
| **Hard Outlier Detection** | | | | |
| Heb | 8.91 | 15.22 | **3.49** | 0.36 |
| Spa | 14.37 | 17.64 | **4.72** | 0.36 |
| Eng | 9.40 | 20.20 | **3.60** | 0.16 |
| **Soft Outlier Detection** | | | | |
| Heb | 56.74 | 56.20 | **49.73** | 26.97 |
| Spa | 61.26 | 56.95 | **52.44** | 27.71 |
| Eng | 58.59 | 59.50 | **50.00** | 26.77 |

(a) Morphological outlier detection

| Lang | Semantic | | | |
|------|--------|-----|---------|---------|
| | Before | Reg | Non-Reg | Skyline |
| **Hard Outlier Detection** | | | | |
| Heb | 74.41 | 49.56 | **84.11** | 85.34 |
| Spa | 70.85 | 44.82 | **86.58** | 86.28 |
| Eng | 77.45 | 41.10 | **88.70** | 87.76 |
| **Soft Outlier Detection** | | | | |
| Heb | 91.45 | 78.43 | **95.13** | 96.06 |
| Spa | 90.13 | 75.99 | **95.73** | 96.53 |
| Eng | 92.27 | 74.36 | **96.3** | 96.94 |

(b) Semantic outlier detection

Table 3: Before and after projection indicator results. While Regressive distorts the embedding space completely, Non-Regressive achieves our goal - reducing the morphological affinity of the space while not harming the lexical semantics encoded.

## 6.4 Results

Table 2 shows the average F1 scores of probes in predicting morphological features for both I²NLP variants. It is apparent that the regressive I²NLP is more effective than the non-regressive in removing morphological properties.

Table 3 shows the results for the indicator task. The morphological outlier detection results in Table 3a show that the non-regressive I²NLP variant is better in removing morphological properties as the performance after the projection is worse both in term of hard and soft detection.[6] Note that when compared to the performance over the original vectors, applying the regressive I²NLP makes it easier to spot the morphological outlier, in terms of hard

---

[6]For soft outlier detection, the average OPP of a guess is 50% so the table points to an almost complete erasure.

outlier detection. This is the opposite of what is expected of a good morphological erasure method.

This picture is reinforced by the results for the semantic outlier detection in Table 3b, showing that the non-regressive variant is also better at preserving the semantic affinities of the space. Its results are impressively similar to those of the skyline, while the regressive variant hinders the performance even comparing to the baseline. This is in stark contradiction to the results of the probes on the same projected spaces, that pointed to the regressive I²NLP variant as the superior one.

## 6.5 Discussion

As with the previous test case, here as well probes and indicators point in polar opposite directions, with probes giving the advantage to the regressive I²NLP while the DEOD indicator showing that the non-regressive is the superior variant. Also similar to the previous test case is the probes' lack of ability to detect non-linearly expressed information. However, unlike the gender bias case, the probes are still doing better than guessing, since even the results for the regressive variant show worse erasure than the skyline. That is to say that the probes were still able to detect residual linearly expressed morphological information.

The indicator task, on the other hand, is much more complex than the indicators used for gender bias detection. It is not restricted to linear separation and it indicates simultaneously on two types of information in the representations: morphological and lexical semantics. The indicator allows us to verify that the non-regressive I²NLP erases better the information it is supposed to erase, while preserving better the information it is supposed to preserve. The indicator's double purpose allows evaluation of more aspects of the space and examine the I²NLP variants not only from the morphological angle, and for this reason we argue in favor of preference to the non-regressive variant.

## 7 Conclusions

In this work we compare probes that do and do not require training an auxiliary classification model, offering *indicator tasks* as a term that covers the latter kind. We show, in two different test cases, that although it is tempting to think of probes and indicators as complementary in understanding the inner structure of embedding spaces, sometimes the methods result in contradictory conclusions.

We claim that the inherent problems with probes and the flexibility of indicators should lead to prioritizing the latter in interpreting the results of property removal procedures. Nevertheless, probes are of course still useful in inspecting embedding spaces, but their results should not be overstated and they should be used only when interested in the specific insights they hold, e.g., the existence of a linear separation in case of linear probes.

Although the design of indicators requires more creativity and result in tasks that are mostly tailored for specific kinds of questions under investigation, we assert that the advantage they give in better understanding the information, that does and does not exist in word representations, is worth the effort. We therefore conclude that indicators should at the very least be presented alongside probes when examining embedding spaces.

## Limitations

The use of indicator tasks, as presented in this paper, solves many problems existing in the current use of probes to query linguistic information from embedded representations. However, there are of course issues that remain unsolved and may require further research.

For example, the interpretability of the absolute results is still unclear with indicators as it is with probes. Suppose we ran a probing experiment or an indicator task and got 87%. Is it good? How could we tell? Could we safely declare the information exists in the representation? (Belinkov, 2022)

Another issue that may arise with the usage of indicator tasks is the lack of a simple recipe for the creation of such tasks. Despite their disadvantages, the design of probing tasks is simple and merely requires picking a target property and a classifier. On the other hand, the design of indicator tasks remains a tougher mission. In this paper we pointed to similarities between vectors as the main source of information for indicator tasks, and indeed from it can be derived many tasks such as clustering and comparison to other sources of similarity. However, the details may require some tailor-made decisions.

In addition, as a conceptual continuation of intrinsic evaluation methods, the indicator DEOD task may suffer from the same problems that haunted intrinsic tasks, mostly with regards of the definition of similarity (Faruqui et al., 2016). However, our indicator is based on the outlier detection task that tended to these issues by examining similarity not in absolute terms but in relation to similarities between other pairs of words (Camacho-Collados and Navigli, 2016). Another problem of intrinsic evaluation is the lack of guaranteed correlation with performance on downstream tasks (Chiu et al., 2016). However, this problem exists for probes as well (Conneau et al., 2018).

Lastly, it is unclear whether indicator tasks in general, and our DEOD task in particular, can detect linear separability in the representations, which is the most common use of probes. For this reason we do not advocate for the abandonment of probes, but rather for limiting their usage to specific goals, like detection of linear separation.

## Acknowledgements

We thank Shauli Ravfogel and an anonymous reviewer (VEsr in openreview) for insightful comments and fruitful discussions. This research was funded by the European Research Council (ERC-StG grant number 677352), the Israeli Ministry of Science and Technology (MOST grant number 3-17992), and the Israeli Innovation Authority (IIA KAMIN grant), for which we are grateful.

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
