# OpenReview forum: "Is Probing All You Need? Indicator Tasks as an Alternative to Probing Embedding Spaces"
_EMNLP/2023/Conference — EMNLP 2023 Findings_

### Official Review · Reviewer_8oKZ · 2023-07-20

**Soundness:** 3

**Excitement:**

3: Ambivalent: It has merits (e.g., it reports state-of-the-art results, the idea is nice), but there are key weaknesses (e.g., it describes incremental work), and it can significantly benefit from another round of revision. However, I won't object to accepting it if my co-reviewers champion it.

**Paper Topic And Main Contributions:**

The authors present the 'indicator tasks' as an alternative to probes, to evaluate embedding spaces. They show that although indicator tasks can be more laborious to be created, they can more properly capture problems such as biases in embeddings spaces, specially after so linear transformation has been already applied on the embeddings.

**Questions For The Authors:**

A) Would it be possible to present some qualitative analysis on the results to understand better why probe is worse than random and the indicators are not that bad in the results of Case Study 1?

B) Could that be done for Case Study 2 as well?

**Reasons To Accept:**

Evaluation of embeddings, specially towards understading bias, is of interest of the NLP community
They proposed an alternative to probes, which presented better results in the paper

**Reasons To Reject:**

The paper could be clearer. Sometimes it is difficult to follow what has been done
Maybe the analysis of the results could benefit with something more visual, such as the illustration of the errors to understand in which cases probes fail to capture gender biases, specially after using RLACE

**Reproducibility:**

3: Could reproduce the results with some difficulty. The settings of parameters are underspecified or subjectively determined; the training/evaluation data are not widely available.

**Reviewer Confidence:**

2: Willing to defend my evaluation, but it is fairly likely that I missed some details, didn't understand some central points, or can't be sure about the novelty of the work.

---

> ### Author Rebuttal · Authors · 2023-08-29
>
> We thank you for your review and for finding this work interesting for the NLP community.
>
>
> Re: “Sometimes it is difficult to follow what has been done Maybe the analysis of the results could benefit with something more visual, such as the illustration of the errors to understand in which cases probes fail to capture gender biases, specially after using RLACE”
>
>
> We will restructure the experimental setup of both test cases to clarify them.
> We will also expand on why we chose these test cases specifically, as well as the choice of languages and analyze results in light of the different languages and the morphological properties separately.
> We will also provide a visualization in an appendix for the gender bias test case. It will be a visualization of gender-associated professions on a graph ranging from male to female biased, before and after applying RLACE.
>
>
> Questions A & B:
> A qualitative analysis is a good idea. We will include an error analysis in the camera ready. We will also look for explicit examples where bias remains in the representations and probes fail to capture it, cases in which these examples will have shared characteristics will be reported in the camera ready.

---

### Official Review · Reviewer_p7Cp · 2023-08-01

**Soundness:** 3

**Excitement:**

4: Strong: This paper deepens the understanding of some phenomenon or lowers the barriers to an existing research direction.

**Paper Topic And Main Contributions:**

The paper proposes the notion of 'indicator tasks' as a contrasting paradigm to 'probing tasks', subsuming all non-trainable tasks for querying embedding spaces.

**Reasons To Accept:**

I particularly like the second use case where the extent of morphological information in word embeddings is investigated. I think it's of value for the EMNLP audience.

**Reasons To Reject:**

l33f: "stark improvement in the performances of any NLP task they have been applied to" --> that statement is wrong and far too general

l72: describe here why you call them indicator tasks early on

l194: I don't find the explanation of why they're called indicator tasks convincing...  - I'd rather extend the term 'probing task' so that it's more specific

l288: discuss early in the paper why you picked those two use cases

l196: "intervention or entanglement of the task itself with the vectors it is supposed to study" -- unpack, not clear how that statement captures what you do

l254: integrate Section 4 in Section 5

Section 5.3: " All in all, this test case demonstrates that indicators were able to expose information that linear probes could not." --> I'd like to see examples here: one case where the bias remains after RLACE (and is detected by the indicators and ignored by the probe) and one where the bias is removed (and either detected/ignored by one of the evaluation methods).

l462: even though I can see where the authors are coming from, 'Iterative Iterative Nullspace Projection' as a term is weird...


**Reproducibility:**

2: Would be hard pressed to reproduce the results. The contribution depends on data that are simply not available outside the author's institution or consortium; not enough details are provided.

**Reviewer Confidence:**

2: Willing to defend my evaluation, but it is fairly likely that I missed some details, didn't understand some central points, or can't be sure about the novelty of the work.

**Typos Grammar Style And Presentation Improvements:**

See above. Plus typos here and there, e.g.,

l17: this kind of tasks --> these kinds of tasks

l303: methods, that incorporate --> remove comma

---

> ### Author Rebuttal · Authors · 2023-08-29
>
> We thank you for your thorough review, typo fixes and rephrasing suggestions. We are glad you saw the added value of our second test case, we agree that examining morphology is an interesting subject in and of itself and is of value to the community.
>
>
> We were also happy to read that most points mentioned under “reasons to reject” are fairly minute weaknesses or rephrasing suggestions that will be addressed in the camera-ready. Below are specific answers to your most significant points:
>
>
> Re: “l194: I don't find the explanation of why they're called indicator tasks convincing... - I'd rather extend the term 'probing task' so that it's more specific”
> Indicators are just an umbrella term for probing methods that do not involve a trainable classifier. It’s mainly used to avoid calling them “probes that do not involve a trainable classifier” as the term “probe” per se is too broad and overloaded with the assumption of a trainable classifier.
> Re: “I'd like to see examples here: one case where the bias remains after RLACE (and is detected by the indicators and ignored by the probe) and one where the bias is removed (and either detected/ignored by one of the evaluation methods).”
> Thanks for this suggestion. It will indeed be very helpful to demonstrate the details of the differences between the probes and the indicators. we will provide such analysis in the camera-ready version.
>
>
> Re: “l288: discuss early in the paper why you picked those two use cases”
> This is a valuable suggestion and we will follow up on it in the camera ready. In brief, we chose the test cases that are the most dissimilar: one deals with a binary property that is of social importance that was heavily discussed in the literature, and the other examines multiple non-binary properties that are linguistic in nature and that the application of erasure methods to them requires innovation.

---

### Official Review · Reviewer_VEsr · 2023-08-02

**Soundness:** 3

**Excitement:**

2: Mediocre: This paper makes marginal contributions (vs non-contemporaneous work), so I would rather not see it in the conference.

**Missing References:**

1. Information-Theoretic Probing for Linguistic Structure, Pimentel et al,. ACL 2020
2. Information-Theoretic Probing with Minimum Description Length, Voita and Titov, EMNLP 2020

**Paper Topic And Main Contributions:**

This paper centers on contrasting probes that necessitate training a classifier with those that solely analyze the properties of the representations without any training. Based on the perplexity of classifier-based probes, which might possess their own inherent blackbox characteristics, the authors assert that such probes do not distinctly elucidate representational features. As an alternative, they introduce the term "indicator tasks" to describe analyses that reveal a property without necessitating model training through the probing task itself. Using concept erasure, the authors present two case studies in which a probe with a linear auxiliary classifier yields conflicting outcomes when compared to one or more indicator tasks. Ultimately, they advocate prioritizing indicator tasks when examining linguistic properties in representations to draw more plausible conclusions.

**Reasons To Accept:**

1. The authors introduce a compelling topic. If complemented with more recent probing experiments, this paper could become an enlightening guide for selecting representation analysis methods.
2. Utilizing a concept erasure technique on a linguistic feature and subsequently comparing the two analytical paths is a clever approach to unveil the contrasting results.

**Reasons To Reject:**

1. This paper seems to face either an issue of overclaiming or lack of experiments. The concern arises from the fact that all case studies in this paper solely compare a simple linear classifier probe with task-tailored indicator tasks. To address this, the paper should either incorporate more complex and recent probing methods that consider their complexity in the results (e.g., Pareto probing or MDL) or temper its claims by focusing solely on examining general probes with **linear** auxiliary classifiers.
2. This paper lacks a comparison between the two analysis methods concerning the model's final output. While it reveals contradictory results between the classifier-based probing task and the indicator task, the key question is how these results relate to the model's practical use. For example, in the gender debiasing section, we observed a major drop in linear probe accuracy when using RLACE, while other indicators showed only a minor decline. However, determining which approach holds greater significance in the output, such as when filling a masked token, remains uncertain.
3. The Probing Conundrum appears to be resolved or at least mitigated by examining the extent to which linguistic features are well-encoded in the representations. Classifier-based probing methods become informative when incorporating model complexity measures (Considering the probing result + the probe's complexity). As highlighted by Pimentel et al. (cf. Missing references 1), the focus should be on the ease of extracting linguistic features from the representations, as seen in certain approaches like Pareto probing or MDL (cf. Missing references 2).

**Reproducibility:**

3: Could reproduce the results with some difficulty. The settings of parameters are underspecified or subjectively determined; the training/evaluation data are not widely available.

**Reviewer Confidence:**

3: Pretty sure, but there's a chance I missed something. Although I have a good feel for this area in general, I did not carefully check the paper's details, e.g., the math, experimental design, or novelty.

**Typos Grammar Style And Presentation Improvements:**

- It would be interesting to observe the impact of I2NLP on each morphological feature, even if presented in the appendix.

---

> ### Author Rebuttal · Authors · 2023-08-29
>
> We thank you for your review, your expertise in this area is apparent.
> We were especially happy to read that you found our analytical approach clever and “an enlightening guide for selecting representation analysis methods”.
>
>
> Rejection reasons #1 and #3 are complementary and advocate for the examination of complex probing models together with sophisticated probing methods. However, as you correctly noticed, and as we explained for control tasks and amnesic probing (l156-l173), these complex methods do not solve the trade offs of probing – they only mitigate them.
>
>
> For this reason, our paper suggests getting rid of the trade offs in probing altogether, by using indicators instead of probes. This is not to say that complex probing methods do not hold value, and future research should certainly pit them against indicators. But our work was aimed at presenting a simple method that solves a complex problem, and we think that its simplicity makes it superior over complex methods that only mitigate the pitfalls of probing.
>
>
> Regarding the move of focus on ease of extraction, we consciously focused on checking the existence of a property in the representations or a lack thereof — rather than assessing the ease of its extraction. This is especially warranted in cases of completely unwanted properties, such as the case of gender bias.
>
>
> All in all, we find this discussion enlightening and we believe that others in the community will benefit from it as well. We therefore hope that you could see this discussion as a strength rather than a weakness of our work. We will of course  elaborate on the points made here in this discussion, together with the mentioned methods, in the camera ready version, and we’d be happy to continue the discussion either anonymously here or acknowledgedly if the paper gets accepted for publication.
>
>
> Regarding the exclusion of experiments over downstream tasks (rejection reason #2), although this is a good idea in general, it was left out of scope of this work since we had no space for it and we found it less important for our main point. The main reason for that was that we were concerned mostly with the existence of the properties in the representations and not their effect on applications. This is also the reason we experimented with erasure methods as test cases.
>
>
> Re: “It would be interesting to observe the impact of I2NLP on each morphological feature, even if presented in the appendix.”
> This is a great idea and we will provide such breakdown in an appendix in the camera-ready.

---

### Meta-Review · Area_Chair_SaC9 · 2023-09-20

**Recommendation:** 3

**Metareview:**

This paper makes an argument for eschewing probing tasks in favor of using behavioral evaluations (“indicator tasks”) without a trained probe component for understanding the linguistic structure of neural models, largely upon the rationale that linear probes show random chance after linear concept erasure, while behavioral evaluations demonstrate that the model still behaves as if it has some knowledge of the linguistic property of interest even after linear concept erasure. The overall claim, as I understand it from the paper and author discussion, is that the simplicity of behavioral evaluations is preferable over changing the probing methodology to add complexity through controls or sample efficiency arguments.

This paper warranted discussion among the reviewers and AC. I think that the methodological problems of understanding unsupervised learned structure are nuanced, and the reviewers seem to believe this work contributes to the discussion thereof. However, I agree with reviewer VEsr that considerable research has been done in this discussion that this work does not engage with properly. The issue seems to be in whether the experiments in this paper demonstrate an issue not already handled by other advancements in analysis methodology.

From my own perspective, I’d ask the authors to consider discussing whether it really is the case that one of probing or indicator tasks must take precedence over the other; they seem just to be testing rather different hypotheses about the model – one about linear extractivity of interesting properties, the other about behavior, so the fact that they lead to differing results isn’t necessarily a huge methodological problem.

---

### Decision · Program_Chairs · 2023-10-07

**Decision:**

Accept-Findings

**Comment:**

This paper makes an argument for eschewing probing tasks in favor of using behavioral evaluations (“indicator tasks”) without a trained probe component for understanding the linguistic structure of neural models, largely upon the rationale that linear probes show random chance after linear concept erasure, while behavioral evaluations demonstrate that the model still behaves as if it has some knowledge of the linguistic property of interest even after linear concept erasure. The overall claim, as I understand it from the paper and author discussion, is that the simplicity of behavioral evaluations is preferable over changing the probing methodology to add complexity through controls or sample efficiency arguments.

This paper warranted discussion among the reviewers and AC. I think that the methodological problems of understanding unsupervised learned structure are nuanced, and the reviewers seem to believe this work contributes to the discussion thereof. However, I agree with reviewer VEsr that considerable research has been done in this discussion that this work does not engage with properly. The issue seems to be in whether the experiments in this paper demonstrate an issue not already handled by other advancements in analysis methodology.

From my own perspective, I’d ask the authors to consider discussing whether it really is the case that one of probing or indicator tasks must take precedence over the other; they seem just to be testing rather different hypotheses about the model – one about linear extractivity of interesting properties, the other about behavior, so the fact that they lead to differing results isn’t necessarily a huge methodological problem.